# Extremely Potent Block of Bacterial Voltage-Gated Sodium Channels by µ-Conotoxin PIIIA

**DOI:** 10.3390/md17090510

**Published:** 2019-08-29

**Authors:** Rocio K. Finol-Urdaneta, Jeffrey R. McArthur, Vyacheslav S. Korkosh, Sun Huang, Denis McMaster, Robert Glavica, Denis B. Tikhonov, Boris S. Zhorov, Robert J. French

**Affiliations:** 1Department of Physiology & Pharmacology, Cumming School of Medicine, University of Calgary, Calgary, AB T2N 4N1, Canada; 2Illawarra Health and Medical Research Institute, University of Wollongong, Wollongong, NSW 2522, Australia; 3Department of Biochemistry, Brandeis University, Waltham, MA 0254-9110, USA; 4I. M. Sechenov Institute of Evolutionary Physiology and Biochemistry, Russian Academy of Sciences, Saint Petersburg 194223, Russia; 5Department of Biochemistry and Biomedical Sciences, McMaster University, Hamilton, ON L8N 4K1, Canada

**Keywords:** µ-conotoxin PIIIA, voltage-gated sodium channels, bacterial sodium channels, prokaryotic sodium channels (NavBacs), eukaryotic sodium channels (Nav1s), voltage- and use-dependent block

## Abstract

µ-Conotoxin PIIIA, in the sub-picomolar, range inhibits the archetypal bacterial sodium channel NaChBac (NavBh) in a voltage- and use-dependent manner. Peptide µ-conotoxins were first recognized as potent components of the venoms of fish-hunting cone snails that selectively inhibit voltage-gated skeletal muscle sodium channels, thus preventing muscle contraction. Intriguingly, computer simulations predicted that PIIIA binds to prokaryotic channel NavAb with much higher affinity than to fish (and other vertebrates) skeletal muscle sodium channel (Nav 1.4). Here, using whole-cell voltage clamp, we demonstrate that PIIIA inhibits NavBac mediated currents even more potently than predicted. From concentration-response data, with [PIIIA] varying more than 6 orders of magnitude (10^−12^ to 10^−5^ M), we estimated an IC_50_ = ~5 pM, maximal block of 0.95 and a Hill coefficient of 0.81 for the inhibition of peak currents. Inhibition was stronger at depolarized holding potentials and was modulated by the frequency and duration of the stimulation pulses. An important feature of the PIIIA action was acceleration of macroscopic inactivation. Docking of PIIIA in a NaChBac (NavBh) model revealed two interconvertible binding modes. In one mode, PIIIA sterically and electrostatically blocks the permeation pathway. In a second mode, apparent stabilization of the inactivated state was achieved by PIIIA binding between P2 helices and trans-membrane S5s from adjacent channel subunits, partially occluding the outer pore. Together, our experimental and computational results suggest that, besides blocking the channel-mediated currents by directly occluding the conducting pathway, PIIIA may also change the relative populations of conducting (activated) and non-conducting (inactivated) states.

## 1. Introduction

µ-Conotoxins (µCTXs) are toxic peptides from venoms of fish-hunting cone snails. After injection of the venom, µCTXs ultimately cause flaccid paralysis by blocking skeletal muscle voltage-gated sodium channels (Navs). These Navs initiate the muscle action potentials that in turn trigger contraction, enabling cone snails to ingest paralysed prey fish. Accidental envenomation of humans can cause fatal respiratory arrest. Molecular targeting of eukaryotic vertebrate Navs by µCTXs was shown in the 1980s by Olivera and collaborators [1,2]. Actions of µCTXs and various other *Conus* peptides have been reviewed extensively, e.g. [3,4,5]. µCTXs from different *Conus* species interact with the pore region of vertebrate Navs and show remarkable ability to discriminate among closely related eukaryotic Nav channel isoforms (Nav1.1–Nav1.9), and for that reason have been explored, highlighting their potential as potential analgesics [6,7,8]. 

Vertebrate Nav1.x α-subunits are formed by a single chain of approximately 2000 amino acids that folds around a 4-fold axis in which the 4 unique functional domains (I–IV) retain structural homology (4-domain-Nav). That is, each domain is composed by six transmembrane helices (S1–6) where S1-4 comprise the voltage-sensing domain (VSD), and the S5-pore loop-S6 conform the pore domain (PD) (Figure 1a, right). Bacterial Navs, in turn, are assembled from four copies of a shorter polypeptide α-subunit, homologous to one vertebrate domain with four individual α-subunits arranged as a tetramer to form a functional ion channel (homotetrameric-Nav, Figure 1a, left) [9,10,11,12].

The PD of vertebrate 4-domain-Navs forms the Na^+^ selectivity filter (SF), which is comprised of the asymmetric “DEKA” motif from domains I-IV. Distinctively, bacterial homotetrameric-Navs have a SF symmetrically contributed by glutamates, “EEEE” motif, from each of the four monomeric α-subunits (Figure 1b) [13]. Our study of µCTX interactions with bacterial Navs were initially motivated by docking simulations by Chung and collaborators [14,15]. Despite remarkable functional homology, the fundamental structural differences between vertebrate 4-domain channels and bacterial homotetrameric-Navs, pose questions as to how a µCTX, presumably evolved to target selectively the asymmetric 4-domain- channel Nav1.4, is able to inhibit symmetric homotetrameric- bacterial Navs with orders of magnitude higher potency, as suggested by computational studies.

Our study demonstrates sub-picomolar µCTX PIIIA inhibition of two different bacterial sodium channels, NaChBac and NavSp1. Electrophysiological measurements, sequence analysis, and computational predictions suggest a rationale for the potent µCTX PIIIA effect on bacterial sodium channels supported on (a) the combination of a fairly homogenous distribution of excess positive charge on the PIIIA surface, and (b) a complementary excess of acidic residues on the pores of homotetrameric- NavBacs. We present two molecular models of PIIIA-bound channels, one illustrating direct pore occlusion by µCTX’s key arginine, and the other, a possible mechanism by which PIIIA appears to induce and/or stabilize an inactivated, non-conducting channel. 

## 2. Results

In the Results, we first describe several experimental observations that outline the similarities and differences of µCTX action on bacterial Nav channels, as compared with the better-studied effects on vertebrate Nav1.x channels.

### 2.1. Extremely High Affinity Block of NaChBac by µ-Conotoxin PIIIA

At saturating concentrations PIIIA blocks ~95% of NaChBac’s peak current, consistent with nearly complete block of single channels by this conotoxin. Although most µCTXs probably cause all-or-none block of their biological targets, there are a number of precedents for incomplete block of unitary currents following replacement of a key basic residues (arginine or lysine), e.g. GIIIA-R13Q and other homologous substitutions including Q, N, A, K, E, D, W, H [16,17,18]. Even for potential biological targets, channel occlusion may be less than complete, e.g. µCTX KIIIA block of neuronal Nav1.2 [6]. PIIIA block of NaChBac (Figure 2b) shows a Hill coefficient that is slightly less than 1 (0.81 ± 0.12); this is consistent with a minority fraction of the channels being bound, but not blocked, by PIIIA (this point will be expanded in the Results, Section 2.7. *Possible Binding Orientations of PIIIA in NaChBac*).

### 2.2. Replacement of a Key “Blocking” Residue (PIIIA-R14A) Abolishes the Speeding of Inactivation and Reduces Apparent Affinity by ~100-Fold, but Does Not Prevent Reduction of Current

To illustrate a complexity of PIIIA interaction with NaChBac that is not seen in experiments with mammalian Nav1.x channels we assessed the effects of PIIIA-R14A on NaChBac mediated currents (Figure 3). Substitution R14A decreases the affinity of PIIIA for mammalian channels and produces incomplete block at the single-channel level [19,20]. 

Figure 3a displays representative examples of NaChBac currents elicited by test pulses to −10 mV (Vh = −100mV, 0.1 Hz). The colored traces correspond to peptide application of 0.1 nM PIIIA wt (red) and 30 nM PIIIA-R14A (orange). Peak current inhibition in both examples is similar, but it is apparent that the key arginine mutant R14A is ~300-fold less potent than the wt peptide. 

Furthermore, Figure 3a includes scaled current traces recorded in the presence of the peptide (in grey). It can be observed that PIIIA-R14A has a less pronounced effect in NaChBac’s inactivation kinetics. Assessment of the activation-linked inactivation kinetics was performed by exponential fits to the current decay during the stimulus pulse and it is presented in Figure 3b as relative inactivation rates to the control (τ_inact_ t/τ_inact_ 0). In this plot it can be seen that unlike wt PIIIA (>90%), PIIIA-R14A yields only a small (~20%) reduction of the rate of inactivation decay. Thus, there appears to be a common requirement of R14 for both pore block, and for speeding of the inactivation decay.

### 2.3. PIIIA Inhibits Navbacs from B. Halodurans and S.pomeroy: Inactivation Is Shifted toward More Negative Voltages, without a Measurable Shift in Activation

In Figure 4 and Table 1, we show that, without any significant shift in activation, prepulse-induced inactivation (or steady state inactivation, SSI) is shifted toward more negative voltages in the presence of PIIIA, for both NaChBac and NavSp1. Whole-cell voltage clamp records measuring unblocked currents in the presence of PIIIA in Figure 4a show a substantial negative shift of about 25 mV in the voltage dependence of prepulse-induced inactivation, for these two NavBacs, which have quite different activation and inactivation kinetics (Figure 4a,c). It is possible that currents during the relatively brief depolarizing test pulses reflect sodium influx through toxin-unbound channels. On the other hand, it is conceivable that are more conformations of toxin-bound NavBacs, e.g., bound-unblocked, bound-slow-inactivated, and bound-blocked, with the first two representing intermediate states on a path to the maximally blocked species (Figure 2 and Figure 3). Thus, the observed voltage shifts of inactivation appear to reflect a direct effect of PIIIA on the inactivation process, rather than resulting indirectly from modulation of the activation process by PIIIA.

### 2.4. Activity- or State- Dependence of µ-Conotoxin PIIIA Inhibition of NaChBac and NavSp1, two NavBac Channels with Substantially Different Kinetics

A novel way in which PIIIA and other µCTXs could reduce NavBac whole-cell conductance is by stabilization of non-conducting (e.g., de-activated or slow inactivated) states during different voltage-activation protocols. Figure 5 illustrates the differential effects of μCTX PIIIA on NaChBac and NavSp1. The traces shown are representative of 4–6 experiments per channel/condition. The slower inactivating current traces from NaChBac in control conditions quickly (washin onset_Inact_ 24.8 ± 12.8 sec, *n* = 5) become ~35-fold faster in the presence of 0.5 pM PIIIA (τ_inact_ Ctr: 177.6 ± 25.9 ms vs τ_inact_ PIIIA: 4.9 ± 1.3 ms, n = 4, *p* = 0.0006) and it is followed by a pronounced decrease in peak currents (washin onset 52.4 ± 17.1 sec, *n* = 5). The diary plots shown in Figure 5b display the relative change in inactivation time constant (assessed by exponential fits to the current decay during the stimulus pulse) and the relative change in peak current from control conditions to PIIIA modified currents. The speeding of inactivation effect caused by PIIIA can be partially reversed (30.5 ± 6.9%) by several minute long washouts, whereas the peak current effect could often be reversed almost completely (79 ± 14%) within 5 minutes of bath exchange (Figure 5a,b). The faster inactivating channel NavSp1 reacts to PIIIA exposure similarly than NaChBac, however, the inactivation time constant is only ~2-fold faster in PIIIA (τ_inact_ Ctr: 9.8 ± 1.9 ms vs τ_inact_ PIIIA: 4.5 ± 1.8 ms, *n* = 4, *p*= 0.0893) and the kinetics of onset of speeding of inactivation and peak current inhibition are the same (onset_Inact_: 201.4 ± 27.1 vs onset_Peak_: 174.7 ± 33.3 sec, *n* = 4, *p* = 0.5513) (Figure 5c,d). This observation suggests a saturating effect on the speeding of inactivation caused by PIIIA on bacterial sodium channels whereby the slower inactivating NaChBac is more evidently affected than the faster NavSp1. 

### 2.5. Holding Potential and Ionic Conditions Affect PIIIA Inhibition of NavBacs. Possible Effects of PIIIA Interactions with Ions in The Pore?

Other experiments show that the fraction of steady-state block at the end of a train of depolarizing pulses increases as the holding potential (Vh) is hyperpolarized in the range −110, −120, −140 mV, presumably because the more negative voltages remove the slow inactivation that accumulates at more positive values of Vh (see Appendix A
Figure A1). In addition, reversing the Na^+^ gradient, which would alter the relative probabilities of the different ions occupying the channel and thus modify PIIIA binding in the pore, also changes the fractional block of the current by PIIIA (see Appendix A
Figure A2).

Thus, Appendix A, Figure A1 and Figure A2, provide further information relevant to conditions that influence use dependence, and effects of ion-toxin interactions within the pore on PIIIA activity. These data also suggest emerging hypotheses, to be tested in more detail in future experiments.

### 2.6. The Slowly Inactivating Mutant NaChBac-G219V is Less Vulnerable to PIIIA Peak Current Block than the Wild Type NaChBac Channel

Functional and computational work from various labs support the hypothesis that glycine residue 219 confers flexibility and acts as a hinge point in the S6 segment of NaChBac channels [21,22,23]. Overall, these studies support the idea that S6-segment mutations at position 219 that enhance kinking of this α-helix stabilize the open conformation [23]. We performed complementary experiments to examine PIIIA action on the NaChBac mutant G219V, which does not display measurable single-pulse inactivation during 10 seconds pulses (data not shown). Figure 6a presents representative traces of NaChBac-G219V mediated currents exposed to 50nM PIIIA. It can be inferred that PIIIA shows about 10,000-fold weaker block (42.6 ± 7.1% block, *n* = 4) compared to wt NaChBac mediated currents (Figure 2). As for the wild type channel, the block of G219V mutant was reversible, with cumulative reduction of peak current (
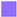
), as well as inactivation-like, single-pulse decay (
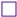
), which became complete within a 300 ms pulse, after ~200 s pulsing at 0.1 Hz (Figure 6b, τ_inact_ PIIIA: 55.9 ± 9.8 ms, *n* = 4). 

Together, these observations and the data presented in Figure 3, show conclusively that mutations in either toxin or channel can decrease the potency of block, but can also modulate the ability of the toxin to accelerate inactivation.

### 2.7. Possible Binding Orientations of PIIIA in NaChBac

Structural aspects of the interactions of PIIIA with mammalian sodium channels are subject of several experimental [19,20] and theoretical [15,18,25,26] studies. However, much less is known about PIIIA interactions with prokaryotic channels. Significant difference in structures of eukaryotic and prokaryotic channels on one hand, and in the electrophysiological characteristics of PIIIA action on these channels on the other hand, necessitate development of specific models, which may or may not be similar to the published structures. For example, Chen and Chung [14] docked PIIIA in NavAb and predicted extra high blocking potency due to toxin interactions with multiple negatively charged residues, c.f. [15]. Here, we studied NavBac channels that, in particular, lack the R62 residue (Figure 1), whose homolog is predicted to contribute to the PIIIA binding site in NavAb [14]. Furthermore, the observed effects of PIIIA and PIIIA-R14A on NavBac inactivation (Figure 3) would benefit from structural rationalization.

Homology modeling was performed as previously described [27]. Docking of PIIIA in the outer NaChBac pore from different starting positions yielded a model (Figure 7a,b), which is conceptually similar to that proposed for PIIIA-bound NavAb [14]. In our model, the side chain of R14 penetrates into the outer pore and interacts with all of the four selectivity filter glutamates. R12 forms a salt bridge with an aspartate in the loop between S5 and P1 helices. In this binding mode, PIIIA covers the entire outer pore and therefore corresponds to the steric and electrostatic pore-blocking mechanism of current inhibition.

Inactivation of bacterial sodium channels, which resembles slow inactivation in mammalian Navs [28], C-type inactivation of potassium channels [29,30,31] and calcium-dependent inactivation of calcium channels [32], all involve gating at the level of the outer pore and the selectivity filter. A recent study demonstrated movement of P1 and P2 helices upon inactivation of a prokaryotic sodium channel [28]. However, the precise, channel-specific conformational changes associated with inactivation are incompletely understood. To illustrate how PIIIA could induce or stabilize an impermeable channel state without completely occluding the pore, we computationally forced the toxin to shift from the pore axis, while preserving R14 contact with one of the selectivity filter glutamates. Under these forces the toxin-filled the groove between adjacent P2 helices and caused reorientation of the selectivity-filter glutamate (E191, cf. Figure 7b,d)). In this binding position, the toxin-channel interactions can cause rearrangements of the P2 helices. Thus, the two binding modes are distinguished by the toxin orientations and various toxin-channel contacts. Furthermore, transition between such modes seems possible without toxin unbinding from the channel. Lacking confirmed experimental data on the structures of inactivated channels, we did not attempt to model such rearrangements. 

## 3. Discussion

Recent structural analysis, simulation/modeling, and experimental studies of both eukaryotic and prokaryotic Nav channels provide a rich background for interpretation of our data (see [9]), and for guided speculation about underlying molecular bases of our observations. Molecular simulations have been reviewed in a variety of functional and evolutionary issues [33,34,35,36,37,38]. Other discussions deal with modulation, and potential roles of tissue- or organ-specific Nav channel gating [39,40,41,42,43,44].

Landmark studies by Nieng Yan and collaborators on eukaryotic and prokaryotic Navs [45,46,47,48,49,50] point to the feasibility of more detailed, specific studies to come, including visualization of channels bound to various ligands, such as highly specific toxins and conventional pharmacological modulators, and may include more detailed evaluation of the functional roles of auxiliary subunits.

### 3.1. Does Extremely High Affinity of PIIIA Result from Complementary Charge Arrays on PIIIA and NaChBac?

This follows intuitively from a relatively large net positive charge calculated at physiological pH [51] and the approximately symmetric distribution of basic residues on the surface of PIIIA, which would complement the near symmetric distribution of acidic residues on the S5-P1 loops of NaChBac and NavSp1, hence providing a strong electrostatic component to their interaction.

Our main experimental observations illustrate exceptionally high affinity block of NaChBac by PIIIA, and outline changes in voltage dependence and kinetics of gating, associated with PIIIA’s presence. The slow onset of PIIIA actions at low concentrations required a train of depolarizing pulses to monitor NaChBac inhibition, while avoiding cumulative inactivation that would result from prolonged inactivation. Qualitatively similar results were observed for a second prokaryotic channel, NavSp1, which shows faster kinetics of activation and inactivation. 

PIIIA action changes following alanine substitution of its key residue R14, which reduces both the affinity and the fractional block of single-channel current for mammalian channels [19,20]. Mutation R14A reduced PIIIA inhibitory affinity for NaChBac, and its action to speed single-pulse inactivation decay.

### 3.2. Gating Modulation (e.g., Enhanced Inactivation) vs. Physical Pore Block

Do simultaneous PIIIA interactions with pore domains from 2 separate subunits underlie the complexities of gating changes for certain NavBacs in the presence of PIIIA? The complex interaction of ligands with channels in different functional states (open, closed, and inactivated by various mechanisms) has been a matter of intensive debates for several decades, stimulated by Hille’s “modulated receptor” hypothesis [52], and developed in parallel by Hondeghem and Katzung for antiarrhythmic drugs [53]. The mechanisms of use-dependent drug action are of fundamental importance for general physiology and drug design. 

Although currently the prokaryotic channels are not actual drug targets, studies of these more simple proteins can help us to understand basic mechanisms of state-dependent actions of different drugs and toxins. The relative simplicity of these channels, i.e. structural symmetry and lack of “fast” inactivation, makes them attractive models to address such questions. Importantly, it is the lack of a fast inactivation (“ball and chain” or “hinged lid mechanism”), which allowed us to reveal interesting peculiarities of PIIIA action on NaChBac, including stabilization/induction of the inactivated state, which is a likely functional analog of the slow-inactivated state in eukaryotic channels. In turn, experimental separation of the pore block and modulation of inactivation has allowed us to suggest novel ideas on the mechanisms of toxin action and visualize them in structural models (Figure 7).

Analogous to our observation of the PIIIA effects on prokaryotic Nav channels, the anesthetic propofol inhibits peak currents and promotes activation-linked inactivation in NaChBac [54]. Complementary chemical and computational analyses suggest that propofol allosterically modulates NaChBac gating by binding to multiple channel sites [55]. However, critical differences in the chemical nature of the modulators (small molecule, propofol vs. peptide, PIIIA), and the extent of the effects on NaChBac inactivation-deficient mutant T220A (explored in the propofol studies) and NaChBac-G219V (assessed in our µCTX work) does not allow a detailed mechanistic comparison of these studies.

It is worth remembering that a cogent argument was made, by Moczydlowski and collaborators that classic guanidinium toxins were unlikely to be inhibiting eukaryotic Navs by direct, physical pore occlusion [56,57]. Although the accumulated data set more than 30 years later, is more complex and extensiove, high affinity pore block remains a cornerstone in understanding inhibition by tetrodotoxin, saxitoxin, and the µCTXs.

### 3.3. Broadening the Scope of µCTX Pharmacology: Homotetramers vs. 4-Domain Channels

Recent studies [58,59] have shown that µCTXs can block mammalian homotetrameric voltage-gated potassium channels (Kvs), but assays were performed with a standard 10 µM test concentration. Although promiscuous action extending to Kv channels was verified, the effective concentrations required were about 10^7^-fold higher than those we have explored for homotetrameric NavBacs. A recent report in Marine Drugs from the same group extends the discussion to fifteen new 3-disulphide-bonded isomers, plus 3 PIIIA mutants in which one of the disulphide bonds was omitted [60]. Much painstaking chemistry and testing will presumably be required to attain the potency close to that observed for NaChBac. Nevertheless, it seems that many interesting mechanistic insights are likely to emerge from further exploration of µCTX targeting promiscuity.

### 3.4. µCTX Pharmacology of Invertebrate Homotetrameric-NavBacs

The study of bacterial ion channels has provided fundamental insights into the structural basis of neuronal signaling; however, the native roles of ion channels in bacteria, in many cases, remain unknown. Prokaryotic, homotetrameric NavBacs are believed to drive flagellar movement in some marine and alkali-philic bacteria [11]. Bacterial Nav channels could provide a source of Na^+^ ions that drives the stators and maintains ion homeostasis, but direct evidence is not currently available [61].

Electrical signaling is commonly viewed as a property of eukaryotic cells, even though cation channels are found in all organisms. Recent findings suggest that bacteria use synchronized oscillations in membrane potential, mediated by K^+^ channels, to coordinate metabolism within biofilms, demonstrating a function for prokaryotic ion channels in active, long-range electrical signaling within cellular communities [62].

Single celled organisms like diatoms exhibit spontaneous action potentials resembling those produced by eukaryotic 4-domain-Navs [63,64]. However, only a few diatom species have 4-domain-Nav-like sequences, while all diatom genomes, identified to date, report on a vast collection of homotetrameric-Nav like sequences akin to the bacterial, homotetrameric-Nav, NaChBac. Even though recombinant NaChBac activates and inactivates significantly slower than mammalian 4-domain-Navs, homotetrameric-Navs found in marine bacteria such as NaSp1 (Figure 4 and Figure 5), have considerably faster kinetics. Helliwell and collaborators have proposed that strongly voltage-gated, fast activating and inactivating, single-domain channels identified in diatom genomes could contribute to membrane excitability and signaling [65]. Thus, NavBac functions in bacterial electrical signaling, and their amenability to modulation by µCTXs, may be a worthwhile avenue for future research.

## 4. Materials and Methods 

General electrophysiological methods and approaches to molecular model building and simulations, as well as kinetic modeling are described in our recent paper of batrachotoxin modulation of bacterial sodium channels [27]. Tikhonov and Zhorov [66] provide additional details of structure prediction based on animal toxin studies. Specific details of application of the µ-conotoxin PIIIA and PIIIA-R14A are provided in the appropriate figures, legends and accompanying text.

### 4.1. Constructs and Mutagenesis

The bacterial sodium channel constructs, NaChBac (*Bacillus halodurans*), and SP1 (*Silicibacter pomeroyi*) were previously described in [27] and provided by D. Clapham (Howard Hughes Medical Institute, Children’s Hospital, and Harvard University, Boston, MA, USA) and D. Minor (Cardiovascular Research Institute, University of California, San Francisco, San Francisco, CA, USA) respectively. The G219V was introduced into the NaChBac plasmid using overlapping primer PCR amplification with the desired nucleotide changes, and completely sequenced.

### 4.2. µ-Conotoxin Synthesis

Synthesis of both native PIIIA and PIIIA-R14A were performed by D. McMaster (Peptide services, University of Calgary) as described previously [19,20]. Briefly, the linear peptide was generated through solid phase peptide synthesis using 9-fluorenylmethoxycarbonyl (Fmoc) chemistry on an Applied Biosystems 431A synthesizer (HBTU/HOBT/DIPEA method). Linear peptide was purified via analytical HPLC followed by oxidative folding under equilibrating conditions (air oxidation in the presence of mercaptoethanol (10 µl in 150 ml) to promote formation of stable disulfide bonding, at 4 °C over 2 to 4 days. Peptides formed a single major peak identifed using analytical HPLC, matching previously determined elution times for both PIIIA and PIIIA-R14A. The crude peptide was then isolated from the acidified reaction mixture using reverse-phase extraction and purified to near homogeneity by HPLC. Identity of the purified peptide was confirmed using quantitative amino acid analysis and my matrix-assisted laser desporption ionization mass spectrometric molecular weight determination. Purified peptide was then lyophilized and dissolved in MilliQ water to a stock concentration of 100 mM.

### 4.3. Electrophysiology

Mammalian TSA201-cells [67] were transiently transfected with the channel cDNA using Lipofectamine 2000 (Invitrogen, Carlsbad, CA, USA). Whole-cell patch-clamp was performed at room temperature (20–22 °C) 18-24 hours post-transfection, using an Axopatch 200B amplifier and digitalized with a Digidata 1322A (Molecular Devices, Sunnyvale, CA, USA). Patch pipettes (Corning 8161 glass, Harvard Apparatus, Cambridge, MA, USA) were pulled using a model P-97 Puller (Sutter Instruments, Novato, CA, USA), fire-polished to a final resistance of 1.5-3 MΩ and filled with an intracellular solution (mM): 105 CsF, 35 CsCl or NaCl, 10 EGTA, 10 HEPES, pH 7.4 with CsOH. External solution contained (mM): 142.5 NaCl, 2 CaCl_2_, 2 MgCl_2_, 10 Glucose, 10 HEPES, pH 7.4 with NaOH. µ-Conotoxins were diluted using external solution to their final desired concentrations. Experiments were performed with 50–60% series compensation on cells containing between 1 to 5 nA of whole-cell currents to maintain adequate voltage control. 

### 4.4. Computational Modeling

Methodology of our homology modeling approach and ligand docking is described, e.g., in [18,27].

### 4.5. Data Analysis

Data analysis was performed using standard software including Clampfit (10.7, Molecular Devices, San Jose, CA, USA) and Igor (6.37, WaveMetrics, Portland, OR, USA) Activation, steady-state inactivation and IV curves were fit as described previously [27]. All summary data is presented as mean ± SEM (n), where n is the number of experimental replicates. Statistical significance was determined using Students *t* test with a *p-value* < 0.05.

## Figures and Tables

**Figure 1 marinedrugs-17-00510-f001:**
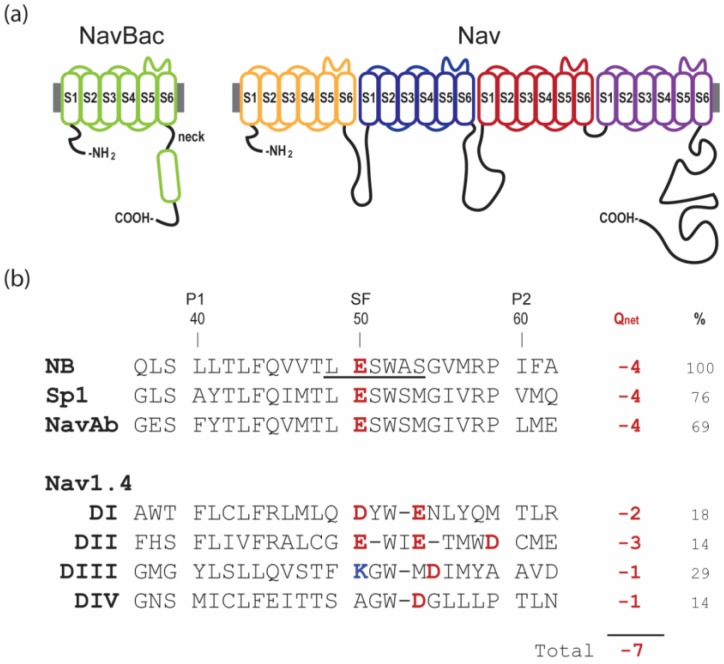
Topology and sequences of voltage-gated sodium channels. (**a**) Topology of bacterial (NavBac) and mammalian (Nav) channels. (**b**) Aligned sequences of the P-loops in prokaryotic and eukaryotic (Nav1.4) voltage-gated sodium channels (pore helix 1, P1; selectivity filter, SF; and pore helix 2, P2). Also shown are % identity, in comparison to NaChBac (NB), and Q_net_, the net charge at the selectivity filter including the inner and outer rings. In Nav1.x, the selectivity filter is formed by the DEKA ring. Notably, not all published alignments of pro- and eukaryotic sequences show the EEEE ring of NavBacs and DEKA ring of Nav1.x channels in matching positions.

**Figure 2 marinedrugs-17-00510-f002:**
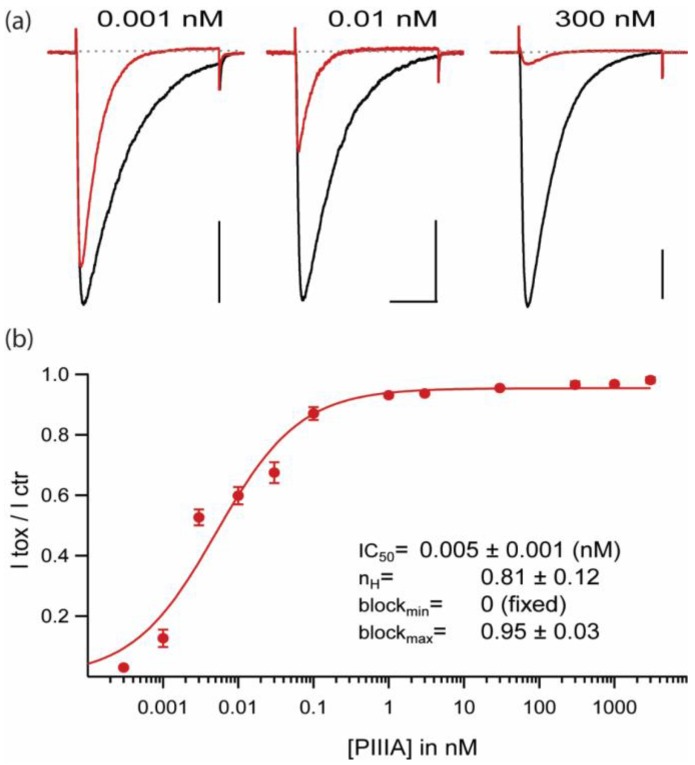
µ-CTX PIIIA blocks NaChBac in the picomolar range. (**a**) NaChBac-mediated currents in control (black) and 3 different PIIIA concentrations (red). Test pulse: −10 mV, 300 ms, from Vh = −100 mV at 0.1 Hz. Scale bars: 0.5 nA, 100 ms. (**b**) Dose-response curve for PIIIA inhibition of peak currents over 8 orders of magnitude of PIIIA concentrations (Vh = −100 mV, 35 mM Na_i_,105 mM Cs_i_/142.5 mM Na_o_; n = 3–6 per concentration).

**Figure 3 marinedrugs-17-00510-f003:**
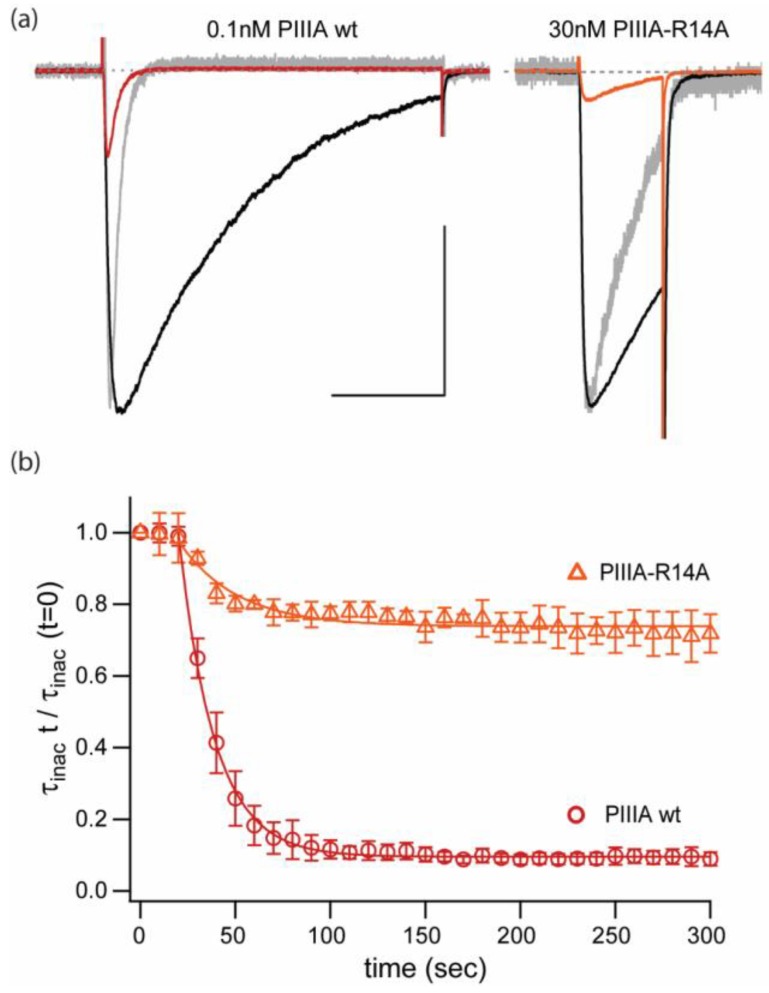
Inhibitory and kinetic effects of PIIIA, and of weakly blocking mutant, PIIIA-R14A. (**a**) Current traces elicited by pulses to −10 mV, from Vh = −100 mV at 0.1 Hz stimulation. Black: control, red: PIIIA, orange: PIIIA-R14A, grey: scaled currents in presence of peptide. Scale bar: 1 nA, 100 ms. (**b**) Relative time constants for inactivation decay normalized to the value at the beginning of the experiment for NaChBac currents inhibited by PIIIA wt (
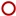
) and PIIIA-R14A (
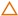
). Data represents mean ± sem from 3 different cells. The R14A substitution appears to affect differentially inhibition/block and speeding of the inactivation rate seen for the wt PIIIA conotoxin.

**Figure 4 marinedrugs-17-00510-f004:**
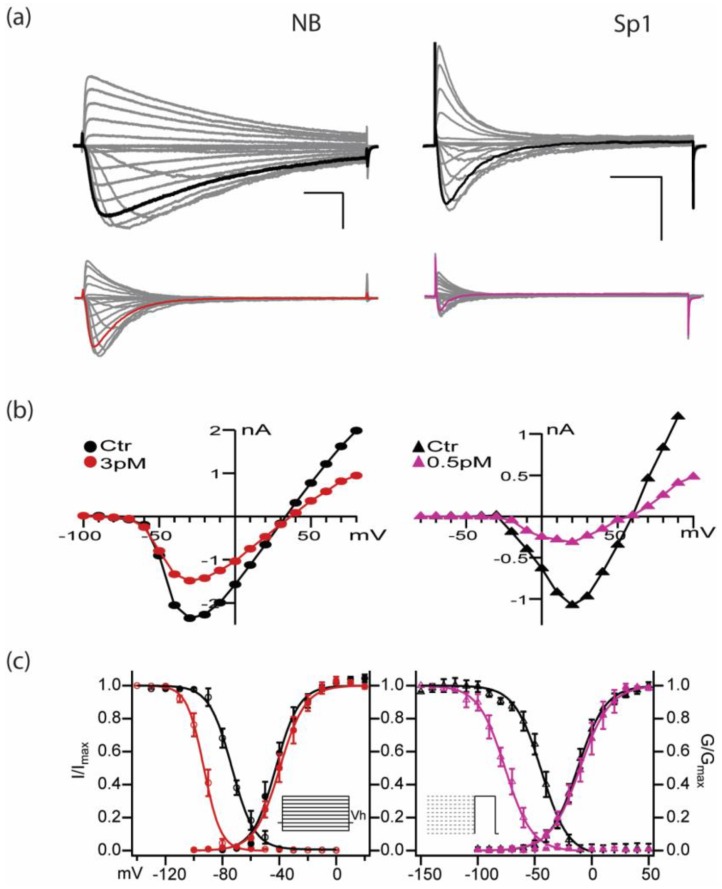
µ-CTX PIIIA inhibits both NB (NaChBac) and Sp1 channels with high potency. (**a**) NB (left) and Sp1 (right) mediated currents in response to IV protocol (Vh= -120 mV, no p/n correction; concentrations in mM 142.5 Na_o_; NB: 35 Na_i_, 105 Cs_i_; Sp1: 140 Cs_i_). Scale bars are 1 nA, 20 ms. Top: control, bottom PIIIA. Currents at -10 mV (NB red) and +30 mV (Sp1 magenta) are highlighted in black (control), red and magenta (PIIIA). (**b**) Peak I-V relationship in control (
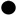
 NB, 
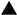
 Sp1) and in the presence of PIIIA for NB (
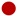
, left) and Sp1 (
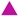
, right). (**c**) Left: NB and Right: Sp1 activation and inactivation plots for control (NB 
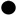
, Sp1 
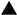
), and in presence of PIIIA (NB 
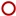
, Sp1 
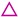
). Data are presented as mean ± sem from 4 replicates per condition, parameters are compiled in Table 1. Protocols are summarized in the insets.

**Figure 5 marinedrugs-17-00510-f005:**
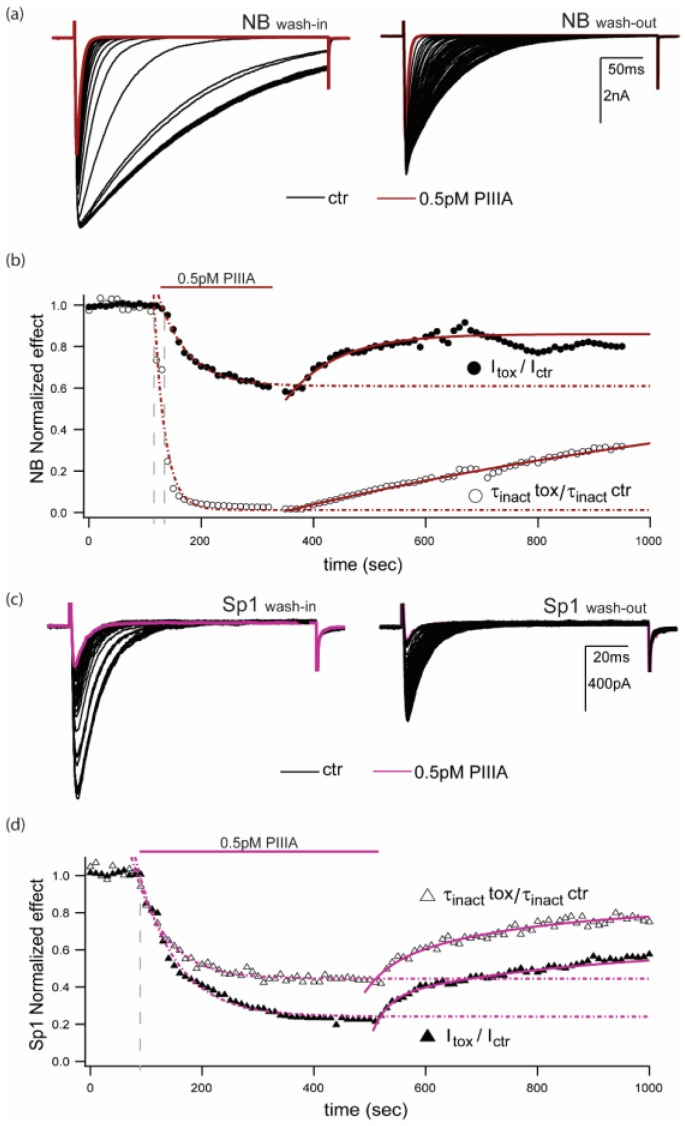
Repeated depolarizations enhance PIIIA block and speed inactivation in NavBacs. (**a**) Representative NaChBac-mediated currents during wash-in and wash-out of 0.5 pM PIIIA (test pulse, Vt = -10 mV; Vh = -120 mV; red trace represents the last point of the wash-in and the beginning of the wash-out). (**b**) Diary plots of the above experiment, showing relative peak currents, and relative inactivation time constants “τ” (normalized to the maximum value for each case); red traces provide the last point of the wash-in, and beginning point of the wash-out. (**c**) Similar plots to those in part (**a**), but from a cell expressing NavSp1 (Vt = 30 mV; Vh= −120 mV; magenta traces represent last point of the wash-in, and beginning of the wash-out). (**d**) Diary plots for NavSp1, as for part (**b**). Data presented here, are representative of 4 experiments per condition.

**Figure 6 marinedrugs-17-00510-f006:**
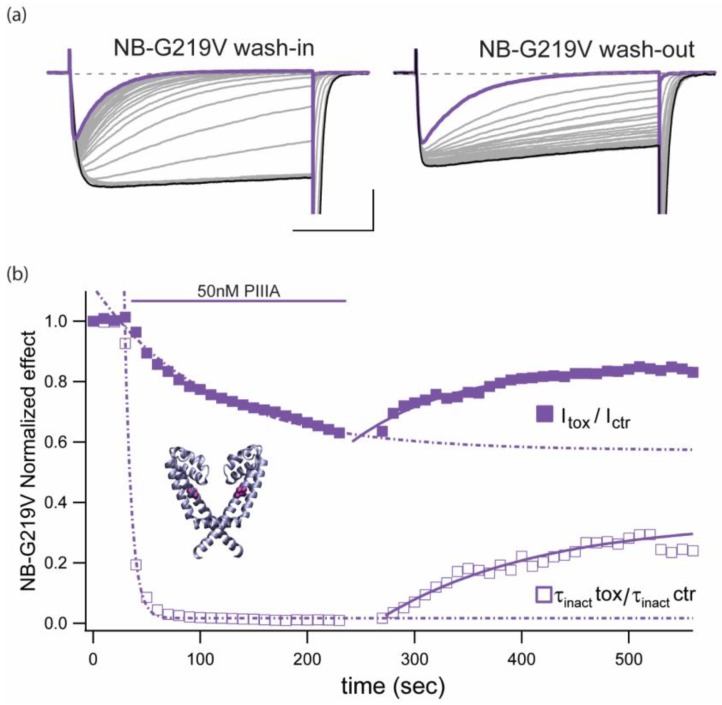
Slowly inactivating NaChBac mutant, NB-G219V, shows enhanced inactivation and decreasing peak current in the presence of μCTX PIIIA. (**a**) NB-G219V-mediated currents during wash-in (left) and wash-out (right) of 50 nM PIIIA (Vt = −10 mV; Vh = −140 mV; 140 mM Cs_i_/142.5 mM Na_o_). Scale bars are 0.5 nA, 100 ms. Purple traces represent the last point of the wash-in and the beginning of the wash-out. (**b**) Diary plot of the experiment shown above including the relative decrease on peak currents and the relative change in inactivation time constant. Data are representative of 3 experiments per condition. Inset shows position of G219 mutation (purple) within the pore domain of the NavMs crystal structure (PDB ID 5HVX; [24]). Peak current for successive pulses decreases with a slower time course than the decrease in τ_inact_. In contrast, on removal of PIIIA, recovery of both τ_inact_ and peak current amplitude followed nearly the same time course.

**Figure 7 marinedrugs-17-00510-f007:**
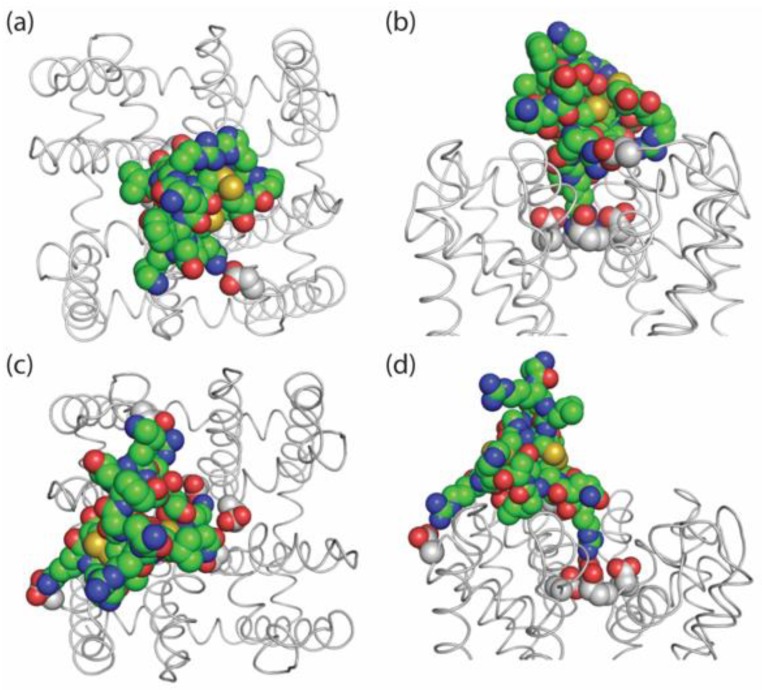
Models of PIIIA docked in NaChBac. Carbon atoms in PIIIA and the channel are green and gray, respectively. Top (**a**) and side (**b**) views of the model with PIIIA blocking the current. Long side chain of R14 penetrates into the outer pore and interacts with the four selectivity-filter glutamates (NB_E191; NavAb_E177), providing both steric and electrostatic block. Other charged residues of PIIIA interact with the P-loops stabilizing the toxin within the channel. PIIIA was docked from different starting positions above the channel. Due to the channel symmetry, very similar toxin orientations were obtained. Top (**c**) and side (**d**) views of the model with PIIIA stabilizing the inactivated channel. The toxin binds between two subunits. R12 and R20 forming salt bridges with the P-loop and S5. R14 interacts with two selectivity filter glutamates, which turn away from the pore axis. This may lead to perturbation of the P1 and P2 helices and stabilization of the inactivated state. Such a binding model is impossible in mammalian sodium channels that have large extracellular loops.

**Table 1 marinedrugs-17-00510-t001:** Parameter values for Figure 4c fits to voltage-dependent activation (G/Gmax vs V), and dependence of channel availability (SSI) to open on holding potential (I/Imax vs Vh).

	NaChBac		Sp1	
	Control	PIIIA	Control	PIIIA
	mean ± sem	mean ± sem	mean = ± sem	mean ± sem
**Activation**				
Gmax	38.8±3.6	22.9 ± 4.4 *	29.7 ± 4.8	12.9 ± 2.20 *
Vhalf, mV	−39.9±2.2	−39.3 ± 3.4	5.6 ± 1.8	5.3 ± 1.44
Slope factor, mV/e-fold	6.7±0.6	8.5 ± 0.7	11.0 ± 1.8	13.3 ± 1.75
**SSI**				
Vhalf, mV	−73.4 ± 4.5	−94 ± 6.7 *	−44.3±2.9	−76.1 ± 5.7 **
Slope factor, mV/e-fold	−7.9 ± 0.9	−5.9 ± 1.4	−11.8±1.1	−10.8 ± 1.2
n	4	4	4	4

* *p* < 0.05; ** *p* < 0.005.

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
