# Peer review of "Extremely Potent Block of Bacterial Voltage-Gated Sodium Channels by µ-Conotoxin PIIIA"

_marinedrugs, 2019, doi:10.3390/md17090510_

Round 1

Reviewer 1 Report

The study assess the inhibition in a bacterial sodium channel (NavBh) by µ-conotoxin PIIIA by electrophsyology and computational modeling. This is an interesting topic in the field. However, the authors in this manuscript didn’t properly design their experiments, didn’t provide sufficient detail of their data collection, has provided insufficient data. A set of data looks wrong, theirfore, the conclusion based on the data will be wrong. Additonally, majorty of the introduction are discussion about the result of the electrophysiology in this study, which is the main part of their experiments. While the discussion is more focus on their computational modeling, a small portion of their experiment.  Overall, this study is far from the finish line. The authors might need to go back to lab, repolish their experiment design, collect more data, analyze their data in a careful and accurate way, do some statistics to support their conclusion, and then revise their paper based on the new data set.

Major comments

1. the authors only provide very limited data as almost all the experiments have only n=1 except figure 2. However, even in figure 2, the authors have not provided how many cells the data were collected from. If the data were collected from multiple cells, please provide this information. But if the data were repeated recording from one cell, it will not be the proper way to plot the figure from the repeated data from a single cell, then did the regression. The n should be at least 3 to be considered as a reliable data set. Based on the data from one cell, it will be questionable to make any comparisons and draw any conclusion.

2. The authors didn’t provide sufficient information in their experiment design, such as the frequency of the repeated recording, what is the duration of pulses, the perfusion time of the toxin before recording, what is the rational of the mutation G219V, what is the rational of applying different holding voltage in different experiment.

3. In the result session, the authors only tested on dose of 30nM PIIA-R14A with a 50 or 60ms pulse test instead of a dose curve, then compared the data to 0.1nM PIIA wt. Without a dose curve of PIIA-R14A, it will be impossible to tell how strong the binding affinity of this compound. For example, if you compare 30nM PIIA-R14A to the same concentration of wt based on the dose curve shown in Fig. 2, you will drawl a wrong conclusion that two compounds show similar inhibition on NaChBac. Additionally, the 50 or 60ms duration of the pulse test to test 30nM PIIA-R14A is too short for the channel to inactivate under control condition. Therefore, the decay time constant calculated based on this will be not accurate, and is not properly to be used to compare to the time constant calculated from the 300ms pulse test of 0.1nM PIIIA wt. Besides this, the data in the supplement Fig. 1c suggested that the duration of the pulse can affect the channel activity, further raising the question about the comparison between the data collected using two different protocols.

4. Addition to #3, 0.1nM PIIA wt only blocked about 75% of the peak current (Fig. 3), which is much less compared to the `85% as shown in figure 2? Can authors give some explanation? Also what is the time point in the decay curve to fit the equation to get the decay time constants? Please provide this information or mark it in your figures.

5. Figure 4a only shows traces of the recording before PIIA treatment. Please provide the traces of the recording after PIIA treatment. Figure 4C is confusion, please clarify which lines are activation curves, and which lines are inactivation curve? The same as the protocols. Also, provide the traces and I-V relationship for inactivation curve experiment. It might be a better way to separate the activation and inactivation curves, such as a, b, c are the traces, I-V relationship, and I/Imax for activation curve. Which d, e, f are the traces, I-V relationship, and I/Imax for activation curve.

6. The activation curve of NB in figure 4c doesn’t match the data shown in Fig 4b, double check the data to make sure the data are correct.

7.  Please mark the time points when the wash-in starts, and when the wash-out start in 5b, 5d, and 6b.

8.  In Fig 6, please provide the unit of scale bar.

9.  The computational modeling only mimic PIIA docked in NaChBac. It will provide more useful information if the authors create models for mutant PIIA, and mutant channels to see if the models fit the electrophysiology data.

10.               In appendix A1a, it looks like there is a time effect or the recording in not stable for all the experiments with different holding voltage (Vh=-120, -140, -110mV). The repeated recording should have a stable baseline as shown in Fig 5 (Vh=-120mV), and Fig 6 (Vh=-140mV). The authors should have this criteria in your method session. Therefore, fig A1a should not be considered as useful data, but a pilot data.

11.               Same question about Fig A1c, it looks like the I/Imax is increasing with time under control regardless of the Vh. The authors might need to have a stable baseline using 300ms pulse duration as shown in the main text, and then test different pulse durations. Without the stable baseline, the data is not properly considered as useful data.

12.               In Fig. A2, it looks like there are wash-out steps or reversing the Na+ gradient based on the context in the result session (line 184 to 186, p6). However, there is no any description in the Figure legend and traces of that step are also missing in Fig. A2a and A2ab. Also provide the method how to reverse the Na+ gradient because if you Cs+ already inside the cell, how can remove it from the inside of the cell and replace it with Na+? Please also provide the duration of the pulse test in Fig. 2.

13.               However, the authors declaimed that they are using t test although there is no statistics in this study. Please correct it. Same as the n number.

Minor comment:

1.                The colors in the same figure are hard to disquintish, for example, purple and black, orange and red. It will be much better to use more disquintish colors.

2.                NavBh is only mentioned in the abstract, while NaChBar is used the main text. It should be consistent through the whole manuscript.

3.                Line 148 in p5, there is a typo error in the title of 2.3. “inactivation is shifter toeard…” should be “Inactivatio is shifted toward….”.

Reviewer 2 Report

This study describes the inhibitory action of PIIIA µconotoxin  on  procariotic sodium channels proposing and discussing two different modes of action.

I have some comments:

In the Abstract remove authors and year from line 22

Introduction

in line 50 quote two important consistent reviews by Tosti et al., and by Ramirez et al. both in Marine Drugs, 2017.

From line 76  to 105,  authors  describe and discuss too many of their methodologies and data. These informations are inappropriate for the introduction, better to move them to the discussion. On the contrary they do not well introduce the scientific background of their experimental work as the impact of conotoxins on different sodium channels subtypes the rationale of these studies, the mechanism of action and the potential therapeutic application of these findings.

In the results from line 222 to the end, authors again discuss some interesting information but do not describe their own results. Therefore this description is inappropriate for results section. So authors must move it in the discussion.

In the discussion, although speculative, it should be of value to add few lines to predict possible biotechnological applications of these findings.

Materials and methods are simple and well designed the contribution of by Drs D. Clapham and  D. Minor are highly appreciated. 

Line 148 typo

This paper is interesting well written and well designed. Authors provide new insights on the mechanism of action of the marine toxins from Conus genus on the ion channels activity by using  prokaryotic models.

I believe that  this paper is consistent with the aim and scope of Marine Drugs and should be of interest for a specialized readership, therefore the MS is suitable for publication after minor revision.

Reviewer 3 Report

This article is very interesting, well written and in my opinion, it should be published soon. It focuses on a topic that has not been thoroughly investigated.

The topic is extremely important and the experiments are well designed. I have few observations:

·         Voltage-gated sodium channels are abbreviated with the Na symbol, followed by a subscript capital V. NaV. (I don’t know if this is visible here). I recommend to change all NaVs and other voltage-gated channels mentioned, like KV.

·         Italics are missing in all scientific names (Conus).

·         In Figure 1, sequences of NaV1.4 are clearly indicated, but NaVBac is not.

·         Lines 71-74. I think this belongs in Results, not in Introduction.

·         You mention Figure 1, Figure 2, then a Figure A1(a) and Figure A1(c), and this is not explained. It is very confusing. Where are these figures? Why?

·         Then you skip to Figure 4. What about Figure 3?

·         Also, figures should be shown after the explanation is given.

·         There is no mention of Figure 3.

·         Then you mention Figure 7 (line 132). After this, Figure 3 is shown with no previous mention in the text. The explanation comes until Line 141.

·         Line 148. I think you meant to write “toward” and not toeard.

·         Lines 166-169. Please clarify this idea. It is difficult to understand because of bad use of commas.

·         Table 1 is never mentioned or explained in the text.

·         Figure 5 is not mentioned in the text.

·         Figure 7. This is very nice.

·         Line 216. After a period you initiate a sentence with parenthesis and then capital T (c and d) Top and side… please check this.

·         Line 254. This is a subtitle and I was expecting more information than to be sent to the Appendixes.

·         Discussion is very well developed; the ideas are clear and enlightening.

·         Lines 306 and 307: The word Thus is repeated and it looks weird.

·         I have no comments on M&M, everything looks fine, except for line 356, where p-value should be lower case and in italics.

·         Maybe figures in Appendixes should be part of the main body of the paper.

·         References: all italics are missing in scientific names.
